# Preparation and Characterization of Intracellular and Exopolysaccharides during Cycle Cultivation of *Spirulina platensis*

**DOI:** 10.3390/foods12051067

**Published:** 2023-03-02

**Authors:** Yuhuan Liu, Yu Wang, Leipeng Cao, Zhenghua Huang, Yue Zhou, Ruijuan Fan, Congmiao Li

**Affiliations:** 1State Key Laboratory of Food Science and Technology, Engineering Research Center for Biomass Conversion, Ministry of Education, Nanchang University, Nanchang 330047, China; 2Research Institute of Quality, Safety and Standards of Agricultural Product, Jiangxi Academy of Agricultural Sciences, Nanchang 330200, China

**Keywords:** *Spirulina platensis*, cultivation, intracellular polysaccharide, extraction, antioxidant activity

## Abstract

The dried cell weight (DCW) of *Spirulina platensis* gradually decreased from 1.52 g/L to 1.18 g/L after five cultivation cycles. Intracellular polysaccharide (IPS) and exopolysaccharide (EPS) content both increased with increased cycle number and duration. IPS content was higher than EPS content. Maximum IPS yield (60.61 mg/g) using thermal high-pressure homogenization was achieved after three homogenization cycles at 60 MPa and an S/I ratio of 1:30. IPS showed a more fibrous, porous, and looser structure, and had a higher glucose content and Mw (272.85 kDa) compared with EPS, which may be indicative of IPS’s higher viscosity and water holding capacity. Although both carbohydrates were acidic, EPS had stronger acidity and thermal stability than IPS; this was accompanied by differences in monosaccharide. IPS exhibited the highest DPPH (EC_50_ = 1.77 mg/mL) and ABTS (EC_50_ = 0.12 mg/mL) radical scavenging capacity, in line with IPS’s higher total phenol content, while simultaneously showing the lowest HO^•^ scavenging and ferrous ion chelating capacities; thus characterizing IPS as a superior antioxidant and EPS as a stronger metal ion chelator.

## 1. Introduction

*Spirulina platensis* is a multicellular filamentous cyanobacterium and has been nicknamed the “Edible Queen” by the FAO and the FDA for its nutritional value [1,2]. *S. platensis* and its derivates have been widely used in dietary supplements and other food products targeted at the health-aware consumer, and are increasingly gaining recognition as functional ingredients [3,4]. Among its most promising derivatives, the polysaccharides of *S. platensis* (PSP) have attracted significant attention due to their antioxidant, antiaging, antiviral, anti-inflammatory, and immunomodulatory potential, as well as their physicochemical attributes [5,6]. Previous studies have shown that PSP is bioactive and comprises polymeric carbohydrates composed of long chains of monosaccharide units bound together by glycosidic linkages, and that its multiple biological activities are closely related to monosaccharide composition, diverse glycosidic linkages, molecular weight, and spatial configuration [7,8]. Moreover, PSP is interesting for its potential to improve intestinal function and health, and to prevent cancer cell proliferation. Therefore, with the increasing demand for PSP in trade markets, the cost-efficient preparation of PSP is currently a pressing research need.

Broadly, PSP can be isolated from the cell bodies of *S. platensis* and from culture media to obtain intracellular polysaccharides (IPSs) and exopolysaccharides (EPSs), respectively. IPSs consist of complex acid sulphate polysaccharides, and account for 15–20% of the cell mass of *S. platensis* cell mass [2,8]. Variations in IPS content are related to culture conditions, including carbon source, light, pH, salinity, and cultivation time. Large-scale extraction of IPS is usually performed by cell disruption, chemical maceration, and enzymatic treatment [9]. On the other hand, EPS is a water-soluble heteropolysaccharide that is secreted during growth and binds tightly to *S. platensis* cell walls, forming a protective capsule against dehydration and toxic agents [10,11,12]. EPS content in *S. platensis* cultures is potentiated by high salinity and low nutrient availability [13]. EPS is obtained by ultrafiltration with membranes intended for the appropriate molecular weight. Both IPS and EPS harbor a variety of functional groups, including -OH, -COOH, -SO_3_H, and -CH_3_, which, together with the structural diversity of these polysaccharides, are thought to give them bioactive properties such as antifungal and antioxidant activity, free-radical scavenging, and inhibition of lipid peroxidation [14,15,16]. However, little is known about how IPS and EPS content may be affected during cycle cultivation, and distinct structural-functional profiles of IPS vs. EPS are yet to be characterized.

Therefore, the aims of the present work are to investigate the influence of *S. platensis* cycle cultivation on IPS and EPS content, and to characterize the structural-functional properties of IPS and EPS; in particular, their physicochemical properties, monosaccharide composition, molecular weight, functional groups, and antioxidant activity. Additionally, an optimized method for the preparation of IPS and EPS based on thermal water coupled with high-pressure homogenization (HPH) is offered. The data herein described thus provides theoretical and technical resources for the cost-efficient production and adequate application of IPS and EPS from *S. platensis*.

## 2. Materials and Methods

### 2.1. Cultivation of S. platensis

*S. platensis* strains (FACHB: GY-D18) were purchased from the Institute of Hydrobiology, Chinese Academy of Science, PR China. Batches of *S. platensis* were cultured in 500 mL Erlenmeyer flasks with 300 mL of modified Zarrouk’s medium for eight days at 25 °C under cycle illumination (5500 lx). After harvesting cells, culture medium was reused to culture a second batch of *S. platensis*, for a total of five cultivation cycles. Deionized water and analytical grade chemicals and solvents were used in all cases. Erlenmeyer flasks and culture medium were sterilized at 121 °C for 20 min before use.

### 2.2. Evaluation of S. platensis Growth

A 20 mL aliquot of cell suspension was filtered through a Whatman filter paper and dried at 105 °C for 24 h. Biomass was calculated according to the method of Zhou et al. [17] and expressed as dry cell weight (DCW; g), following the formula:DCW (g/L) = (W_n_ − W_0_)/0.02
Growth rate (g/L/d) = (DCW_n+1_ − DCW_n_)/1
where W_n_ corresponds to total dry weight (g) of the filter paper with algae, W_0_ represents the dry weight (g) of the filter paper alone, 0.02 is the aliquoted volume (L), and n is time (days).

### 2.3. Experimental Design of RSM for Extraction of IPS from S. platensis 

IPSs were extracted from *S. platensis* cell bodies using hot water coupled with high-pressure homogenization (HPH; GJJ-0.06/100, Shanghai Taichi Light Industry Equipment Co, LTD), according to the experimental design of RSM (Table 1). Extracts were centrifuged at 7104× *g* for 10 min, and supernatants were concentrated at 80 °C. Concentrated supernatants were treated five times with savage reagent (Chloroform: n-butanol = 4:1) to remove protein, then mixed (1:4) with 95% ethanol and allowed to precipitate for 12 h at 4 °C. Precipitates were suspended in a small aliquot of deionized water before dialysis, and then dialyzed in molecular weight cut-off bags (8–10 kDa) for 48 h to eliminate residual salts. Finally, samples were lyophilized, weighed, and kept for further experiments. The yield of IPS (mg/g) was calculated as follows:Yield (mg/g) = c × n × v / m
where c is the concentration (mg/mL), n is the dilution factor, v is the sample volume (mL), and m is the dried sample weight (g).

### 2.4. Extraction of Extracellular Polysaccharide (EPS)

EPSs were obtained from *S. platensis* culture medium according to the method of Li et al. [13]. The medium was filtered through a 0.45 μm membrane, and the filtrate was concentrated at 50 °C and then precipitated with ice-cold ethanol (1:4 *v/v*) at 4 °C for 12 h. Precipitates were dialyzed, lyophilized, weighed, and stored in the same manner as IPS extracts. 

### 2.5. Chemical Composition Analysis of IPS and EPS

Total sugar and protein content were determined by the phenol-sulfuric acid and the Coomassie brilliant blue methods, respectively [18]. Total phenol content was estimated using the Folin–Ciocalteu reagent and measuring absorbance at 750 nm; gallic acid (20–100 µg/mL) served as standard, based on Chaiklahan et al. [8]. Phenol content was expressed in gallic acid equivalents. Ash content was determined based on weight loss after 4 h at 550 °C.

The monosaccharide composition of IPS and EPS was determined with a precolumn derivation HPLC method using 1–phenyl–3-methyl-5–pyrazalone (PMP) (Ma et al., 2019). Samples were thoroughly hydrolyzed to monosaccharides by treatment with 4 M trifluoroacetic acid for 8 h at 110 °C, and then mixed with the PMP solution and chloroform. Samples were then analyzed by HPLC (Agilent, Santa Clara, CA, USA) and UV detection at 245 nm. A 4:21 (*v*/*v*) mixture of acetonitrile and 0.125 mol/L KH_2_PO_4_ was used as the mobile phase at a flow rate of 0.8 mL/min, at 30 °C.

The molecular weight (Mw) of IPS and EPS was determined by gel permeation chromatography (GPC, ELEOS System, Wyatt Technology Co., Goleta, CA, USA), based on the method of Zhang et al. [10] with slightly modified chromatographic conditions: 0.2 mol/L NaNO_3_ served as mobile phase at a flow rate of 0.5 mL/min and a column temperature of 25 °C; injection volume was 20 μL.

### 2.6. Fourier-Transform Infrared (FTIR) Spectroscopy and Scanning Electron Microscopy (SEM)

Infrared spectra of IPS and EPS were obtained using a FTIR spectrometer (Thermo Scientific Nicolet IS50, MA, USA) according to the method of Sasaki et al. [19]. A dried 1.0 mg sample was ground and pressed into tablets mixed with 100 mg of KBr. Tablets were scanned at a wavelength range of 4000-400 cm^−1^.

The surface morphology of IPS and EPS was observed by SEM. The powdered sample was sprinkled on the surface of a piece of double-sided tape which was adhered to the microscope’s aluminum column, and then sputter-coated with platinum powder using an ion sputter coater for observation.

### 2.7. Zeta Potentials and Thermal-Gravimetric (TG) Analysis

Sample solutions (1.0 mg/mL) were prepared in ultrapure water. Zeta potentials were determined at 25 °C in the pH range of 2.0–9.0 using a Zeta sizer Nano-ZS particle diameter and potentiometric analyzer (Malvern Instruments, MC, UK). All samples were measured in triplicate.

The thermodynamic characteristics of IPS and EPS samples were analyzed by differential scanning calorimetry (DSC) (Netzsch, DSC 214 Polyma, Selb, Germany). A 5.0 mg sample was weighed in an aluminum pan, using an empty pan as reference. Measurements were performed under nitrogen flow (40 mL/min), at a heating rate of 10 °C/min in a range of 30 °C to 800 °C.

### 2.8. In Vitro Antioxidant Activity Measurements

#### 2.8.1. DPPH Radical Scavenging Assay

The scavenging activity of different concentrations of IPS and EPS on 2,2-diphenyl-1-picrylhydrazyl (DPPH) radicals was evaluated according to the method of Su et al. [20], with slight modifications. Briefly, 1.0 mL of polysaccharide extract (0–2.5 mg/mL) was thoroughly mixed with 1.0 mL of DPPH solution (0.2 mmol/L in 95% ethanol). The mixture was allowed to react for 30 min, protected from light, and absorbance was then measured at 517 nm with a UV/Vis spectrophotometer. In this step, 95% Ethanol and 0–2.5 mg/mL ascorbic acid were used as blank and positive control, respectively. DPPH radical scavenging activity was calculated with the formula:DPPH radical scavenging ability (%) = (A_0_ − A_1_ + A_2_) × 100/A_0_
where A_0_ represents the absorbance of the DPPH solution alone, A_1_ is the absorbance of the DPPH solution containing the sample, and A_2_ is the absorbance of the ethanol solution with the sample.

#### 2.8.2. ABTS Radical Scavenging Assay

Scavenging activity on 2,2′-Azino-bis (3-ethylbenzthiazoline-6-sulfonate) (ABTS) radicals was analyzed as described by Tian et al. [6], with some modifications. Equal volumes of an aqueous solution of 7.0 mmol/L ABTS and 2.45 mmol/L K_2_S_2_O_8_ were mixed and allowed to incubate at RT for 12 h, while protected from light, to acquire the ABTS+• solution. This ABTS radical solution was then diluted with phosphate buffer solution (pH 7.4) until reaching an absorbance of 0.70 ± 0.02 at 734 nm. A 100 μL aliquot of the diluted ABTS+• solution was mixed with 100 μL of sample (0.1–2.5 mg/mL), and absorbance was measured at 734 nm after 30 s oscillation. Deionized water and ascorbic acid (0–2.5 mg/mL) served as blank and positive control, respectively. ABTS radical scavenging activity was calculated as follows:ABTS radical scavenging ability (%) = (A_0_ − A_1_ + A_2_) × 100/A_0_
where A_0_ is the absorbance of the diluted ABTS+• solution alone, A_1_ is the absorbance of the diluted ABTS+• solution mixed with the sample, and A_2_ stands for the absorbance of the sample in deionized water.

#### 2.8.3. HO• Radical Scavenging Assay

HO• scavenging activity was assayed according to the method described by Ji et al. [21] with some modifications as follows: A 1.0 mL aliquot of a sample solution (0–2.5 mg/mL) in 95% ethanol was thoroughly mixed with 1.0 mL of each of the following: 9 mmol/L H_2_O_2_, 9 mmol/L FeSO_4_, and 9 mmol/L salicylic acid. The solution was then incubated at 37 °C for 60 min with cycle-shaking, and absorbance was measured at 510 nm. Ascorbic acid (0–2.5 mg/mL) was used as a positive control. Hydroxyl radical scavenging activity was calculated as follows:HO• scavenging ability (%) = (A_0_ − A_1_ + A_2_) × 100/A_0_
where A_0_ is the absorbance of deionized water, A_1_ is the absorbance of the sample, and A_2_ is the absorbance of the solution without sample.

#### 2.8.4. Fe^2+^ Chelating Ability

Fe^2+^ chelating ability was determined as described by Chang et al. [22], with minor modifications. Briefly, 1.0 mL of sample (0–2.5 mg/mL) was mixed with 3.7 mL deionized water and 0.1 mL of 2.0 mmol/L FeCl_2_·6H_2_O solution, vigorously stirred for 30 s, and then 0.2 mL of 5 mmol/L ferrozine solution was added. The mixture was incubated for 10 min at 25 °C and absorbance was measured at 562 nm. Deionized water and sodium ethylenediamine tetra acetic acid (EDTA-Na_2_) (0–2.5 mg/mL) were used as blank and positive control, respectively. Chelating capacity (%) was calculated with the formula:Chelating ability (%) = (A_0_ − A_1_) × 100/A_0_
where A_0_ is the absorbance of the reaction solution without sample and A_1_ is the absorbance of the reaction solution with the sample.

#### 2.8.5. EC_50_ Calculation

EC_50_ represents the mass concentration of the sample when clearance is 50%. To calculate EC_50_ values for DPPH, ABTS, and HO• radical scavenging activity, and for Fe^2+^ chelating capacity, the clearance ratios of different sample concentrations were plotted and fitted linearly.

### 2.9. Statistical Analysis

All the experiments were conducted in triplicate. Data plotting was performed with Design Expert 13, Origin 2021, and IBM SPSS Statistics 26. Analysis of variance (ANOVA) was carried out wherever applicable and *p* < 0.01 was regarded as a significant difference. For all figures and tables, data were presented as mean ± std (n = 3) of the three independent replicates.

## 3. Results and Discussion

### 3.1. Change of S. platensis and Polysaccharide Content during Cycle Cultivation

Figure 1 shows the growth curve of *S. platensis* and the growth in polysaccharide content (IPS and EPS) during cycle cultivation. DCW of *S. platensis* and polysaccharide content showed a linear increase with prolonged cultivation time. The growth of *S. platensis* behaved as a parabola, reaching its maximum rate (0.24 g/L/day) on the fourth day of cultivation. After eight days in culture, the DCW of *S. platensis* reached 1.52 g/L, representing a 660% increase, and appeared as a regular spiral filament under the microscope (Figure 1a). Total polysaccharide content significantly increased with extended cultivation time, with IPS (80.08 mg/L) reaching a concentration three times higher than that of EPS (27.94 mg/L) by the end of cultivation (Figure 1b).

As shown in Figure 1c, the DCW of *S. platensis* gradually decreased from 1.52 g/L to 1.18 g/L with each cultivation cycle. This is likely explained by a decrease in the microalgae photosystem II, as a consequence of the accumulation of dissolved organic matter (DOM) and increased viscosity of the culture medium [13]. IPS content increased together with the number of cycles, which may also be due to DOM accumulation and reduced DCWP, reaching 203.34 mg/L (or a 136% increase) after five cultivation cycles. Meanwhile, EPS content increased to 52.62 mg/L by the second cycle and remained stable in subsequent cycles. Reusing culture media several times is likely to curb the availability of nitrogen and other nutrients, which can in turn contribute to increasing the C/N ratio and thus promote the incorporation of carbon into the EPS fraction [11].

### 3.2. Single-Factor Test of IPS Extraction

With the gradual increase in the solid–liquid ratio, the extraction rate was the first to increase and then decrease (Figure 2a), and when the material–liquid ratio was 1:30 g/mL, the extraction rate could reach 54.30 ± 0.75 mg/g. With the increase in high-pressure homogenization pressure, the extraction rate of intracellular polysaccharide reached the maximum at 60 MPa (Figure 2b), and the extraction rate was 48.13 ± 0.90 mg/g. The highest extraction rate was achieved when the number of extractions was three (Figure 2c), and it was 48.40 ± 0.29 mg/g.

### 3.3. Optimization of IPS Extraction

To optimize the extraction procedure of IPS from *S. platensis*, a total of 17 experiments with three independent variables (A = S/I ratio; B = pressure; C = number of homogenizations) were performed following a Box–Behnken design (BBD) (Table 1). IPS yield ranged from 39.87 to 60.33 mg/g (dry weight) across all 17 experiments. Based on multiple regression analysis on the experimental data, a second-order polynomial equation expressing the relationship between each variable was generated:Yield (%) = −161.98 + 4.97A + 3.17B + 33.46C − 0.11AB + 0.06AC − 0.08BC − 0.07A^2^ − 0.02B^2^ − 5.19C^2^

The results and RSM analysis are presented in Table 2. The F value for the model was 363.16 (*p* < 0.0001), indicating that the model was statistically significant. The *p* value of the linear (A; B; C), interaction (AB; BC), and quadratic term coefficients (A^2^; B^2^; C^2^) were all lower than 0.01, which implied that these variables had significant effects on the extraction yield. The correlation coefficient (R^2^) was 0.9979, indicating that the predicted and observed values were similar and that the model was a good fit. In addition, the determination coefficient (R^2^_adj_) was 0.9951, which indicated that only 0.49% of the total variation could not be captured by the regression model. The *p* value for lack of fit was 0.1532, which means that lack of fit and pure error were not significantly different. These results thus indicated that the regression model could adequately predict IPS extraction yield.

The relationship between independent and response variables and response is visually represented as a 3D surface response (Figure 3a). For S/I ratio and pressure, the projection of 3D response surface at the bottom was elliptical, indicating that the mutual interaction between S/I ratio and pressure was significant. A similar trend was observed for S/I ratio and number of homogenizations, and for pressure and number of homogenizations (Figure 3b,c). The peak point at their response surfaces also simultaneously existed in their minimum elliptical, indicating that there was an extremum value in the chosen range.

Based on multiple regression and 3D surface response analyses, the optimal conditions for IPS extraction were predicted as follows: S/I ratio = 1:30.79; pressure = 61.08 MPa; and three homogenizations, for an extraction yield of 60.20 mg/g. A verification experiment was performed under the optimal conditions predicted by the model (S/I ratio = 1:30; pressure = 60 MPa; and three homogenizations). The observed IPS extraction yield was 60.61 mg/g, which was not statistically different from the predicated value. Therefore, the regression model was suitable for the prediction of IPS extraction from *S. platensis*. 

### 3.4. IPS and EPS Composition

Table 3 shows the chemical compositions of IPS and EPS. Both IPS and EPS contained more than 65% total sugars and less than 5% protein. The phenolic content in IPS (7.3%) was higher than in EPS, indicating a stronger antioxidant capacity. The carbohydrates present in both IPS and EPS but in different ratios (Figure 4) included mannitol, ribose, rhamnose, glucuronic acid, galacturonic acid, glucose, galactose, xylose, and fucose. However, the proportion of each monosaccharide content was statistically significantly different in IPS vs. EPS (Table 3). IPS’s main monosaccharides were glucose (83.62%), rhamnose (4.42%), fucose (3.25%), and glucuronic acid (2.39%). In comparison, EPS contained mainly fucose (19.99%), rhamnose (15.61%), glucose (14.75%), galacturonic acid (11.13%), and galactose (10.78%) content, and had a lower molecular weight (185.13 kDa). This indicated that IPS may have a higher viscosity than EPS [19]. These differences in monosaccharide content and Mw between PSP fractions point to remarkably distinct functional properties and potential when used as food additives e.g., as thickening stabilizers.

### 3.5. FTIR Spectrum Analysis and SEM Imaging

The FTIR spectra of IPS and EPS indicated large similarities in the functional groups contained in both polysaccharide fractions (Figure 5a). The absorption peaks observed at around 3413 and 2925 cm^−1^ are typical of the O−H and C−H stretching vibrations in rhamnose and fucose, respectively [23]. The amide I band at 1650 cm^−1^ can be taken to represent the symmetrical and asymmetrical stretching vibration of C=O in COO− and −NHCOCH_3_, together with the bending vibration in the N-H bond [24]. Similarly, the amide II band with peak absorption at 1542 cm^−1^ can be mainly attributed to the symmetrical stretching vibration in the C−O bond. Next, absorption peaks at 1400–1200 cm^−1^ represent variation angle vibrations. The absorption peak at 1240 cm^−1^ can be attributed to the asymmetrical stretching vibration in −S=O, indicating that both IPS and EPS contained a small number of -SO_3_H groups [6]. The presence of the pyran ring and the carbohydrate skeleton (C−O−C) is indicated by their characteristic peaks at 1153 and 1064 cm^−1^, respectively [10]. Finally, the absorption peaks at 898 and 819 cm^−1^ correspond to the deformation mode of the β−D−pyranoside bond (C−H) and α−Mannitose, respectively [25].

To better understand their physical properties, the surface and microstructure of IPS and EPS was visualized by SEM. IPS and EPS were remarkably different in shape and size (Figure 5b). IPS presented a smooth surface with irregular thin stripes at 2000× magnifications. At 5000× magnifications, IPS exhibited a loose, finely lamellar, and porous web-like structure; these characteristics could imply an enhanced solubility exposure of active groups in IPS. In contrast, EPS had a relatively smoother and flatter surface, and a more coarsely lamellar, less porous structure. Because of its fibrous and porous structure, IPS has likely more versatile application in various foods, and may be especially superior for its water holding capacity compared with EPS [10].

### 3.6. Zeta Potential and TG Analysis

The changes in zeta potential of IPS and EPS solutions in response to pH are shown in Figure 6a. As pH increased from 2.0 to 9.0, the zeta potentials of IPS and EPS decreased from −25.73 to −29.77, and from −26.43 to −37.5, respectively. The smaller differential between IPS and EPS in this pH range may be explained by the high glucose content in IPS. Although both extracts had negative zeta potentials, meaning both of them are acidic polysaccharides, EPS showed a more negative potential than IPS overall. This points to EPS’s stronger acidity which, in agreement with previous reports, is probably due to a higher abundance of −SO_3_H in EPS.

Thermal stability is a crucial physicochemical property for the commercial application of polysaccharides. The TG and derivative TG curves were experimentally determined for IPS and EPS (Figure 6b). Analysis of weight loss revealed three major stages: (1) 50–200 °C; (2) 200–500 °C; and (3) 500–800 °C. Weight loss during the first stage was 4.78% for IPS and 10.11% for EPS, and could be attributed to the evaporation and dehydration of adsorbed and surface water from the polysaccharide’s surface. During the second stage, weight loss was approximately 58.47% (IPS) and 53.03% (EPS), and was possibly due to the degradation of long carbohydrate chains and the depolymerization of fragments. By the third stage, weight loss slowed down, only decreasing by 29.94% (IPS) and 13.71% (EPS), which could also be due to the fact that the remaining compounds were further carbonized and some carbonates were converted into CO_2_. Maximal weight loss reached 93.84% (IPS) and 77.36% (EPS) at 800 °C. These results showed that IPS and EPS are thermally stable below 220 °C, and that EPS’s thermal stability is higher, possibly as a consequence of its higher fucose and rhamnose content.

### 3.7. Antioxidant Capacity Analysis

DPPH is a stable nitrogen−centered radical and is widely used for the in vitro evaluation of the antioxidant capacity of natural products [10]. IPS and EPS showed an overall strong scavenging activity on DPPH radicals (Figure 7a), and this was dependent on the concentration of polysaccharide, reaching its maximum value at 2.5 mg/mL (65.9% and 44.7% for IPS and EPS, respectively). The EC_50_ value of IPS (1.77 mg/mL) was lower than that of EPS (4.67 mg/mL). The greater ability to scavenge DPPH radicals of IPS is consistent with its higher phenolic content. For comparison, the DPPH scavenging activities of *S. platensis*−derived polysaccharides are superior to those derived from other bacteria and microalgae, specifically *Pseudomonas fluorescens* (approximately 30% at 1.0 mg/mL EPS) [26] and *Sargassum carpophyllum* (66.6% at 12 mg/mL IPS) [6].

Scavenging of ABTS radicals is another common indicator of the antioxidant potential of natural compounds. Both IPS and EPS were shown to be strongly capable of scavenging ABTS radicals in a concentration-dependent manner (Figure 7b). Scavenging activity reached 95.26% at 1.0 mg/mL and 94.47% at 2.5 mg/mL for IPS and EPS, respectively, and these values were not statistically different from the positive control at the same concentration. The EC_50_ values for ABTS radical scavenging were 0.12 mg/mL (IPS) and 0.60 mg/mL (EPS); this heightened scavenging activity for IPS may be explained by a lower sulphate/sugar content (*p* < 0.05). Likewise, PSP extracts showed better ABTS radical scavenging performance when compared with polysaccharides sourced from *Oudemansiella radicata* mushroom (EC_50_ = at 0.2 mg/mL ORP) [25] and *Botryococcus braunii* (EC_50_ = 5.13 mg/mL EPS) [27].

HO• is a highly reactive radical known for its deleterious biological effects, including red blood cell death, DNA damage, and cell membrane degradation, and is prominently implicated in ageing [28]. For this reason, scavenging HO• radicals constitutes an important antioxidant defense mechanism. Both IPS and EPS presented scavenging activity on HO• radicals and this also increased with concentration (Figure 7c). The scavenging capacity of PSPs is directly related to the function of electrons and hydrogen, as supported by previous reports [6]. The EC_50_ of IPS (1.72 mg/mL) was higher than that of EPS (0.75 mg/mL), *p* < 0.05; this superior ability of EPS to scavenge HO• radicals may stem from its rich alcohol hydroxyl groups in the structure of fucose.

The chelating ability of IPS and EPS on ferrous ions also increased at higher concentrations (Figure 7d). EPS had the strongest chelating capacity (85.91%) at 1.0 mg/mL. Remarkably, this was higher than the positive control’s (73.90%) and IPS’s (40.84%) chelating abilities at same concentration. The EC_50_ of IPS and EPS were 1.54 mg/mL and 0.38 mg/mL, respectively. Thus, EPS showed a stronger chelating power on ferrous ions, which is probably due to the abundance of COO− and SO_4_^2-^ in EPS.

## 4. Conclusions

In these results, the content and functional properties of IPS and EPS were investigated during cycle cultivation of *S. platensis*. The results showed that the DCW of *S. platensis* gradually decreased with the increase in number of cycles during cycle cultivation, and the IPS and EPS content gradually increased with the increase in number of cycles and extension of time during cycle cultivation, and IPS content was far higher than EPS. The maximum yield of IPS (60.61 mg/g) could be obtained under the condition of 1:30 S/I ratio and 60 MPa, three times, using thermal-HPH technology. The same carbohydrates were present in both IPS and EPS but in different ratios. IPS has more loose fibrous porous structures, higher glucose, and larger Mw than EPS, indicating higher water holding capacity and viscosity. Both IPS and EPS were shown to be acidic carbohydrates, but the acidity and thermal stability of EPS were stronger than those of IPS, which might be closely related to the monosaccharide content. IPS exhibited a better scavenging capacity on DPPH and ABTS radicals than EPS, possibly due to higher total phenol content, and far lower scavenging ability on OH• radicals and lower ferrous ion chelating ability than EPS, which indicated that IPS showed high antioxidant capacity, but EPS had strong chelating ability on metal ions. These results could provide theoretical direction for the cost-efficient production and adequate application of IPS and EPS from *S. platensis* as food additives or medicinal ingredients. In future studies, the extraction efficiency of IPS should be improved for large-scale production. Moreover, the rheology, water holding capacity and number of major functional groups of IPS and EPS should also be further analyzed to better understand their functional properties.

## Figures and Tables

**Figure 1 foods-12-01067-f001:**
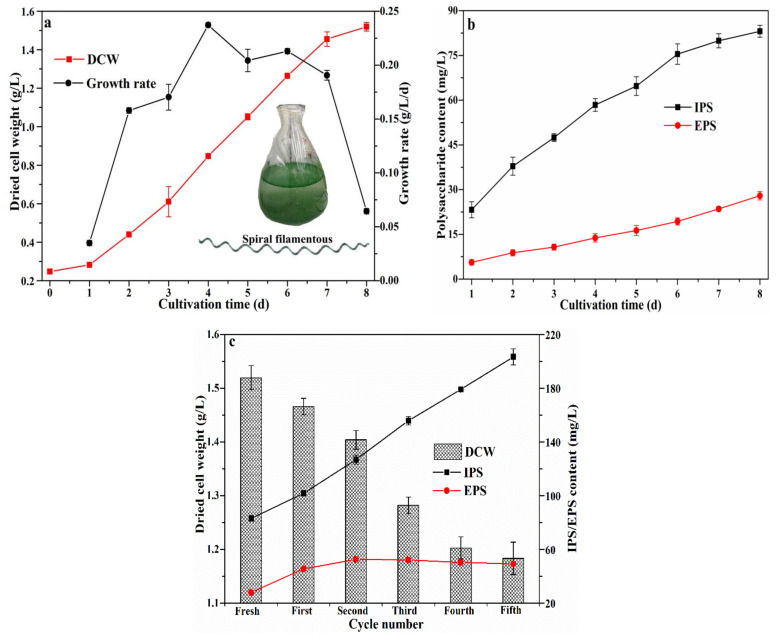
Growth of *Spirulina platensis* (**a**) and change of IPS/EPS (**b**,**c**) during cycle cultivation.

**Figure 2 foods-12-01067-f002:**
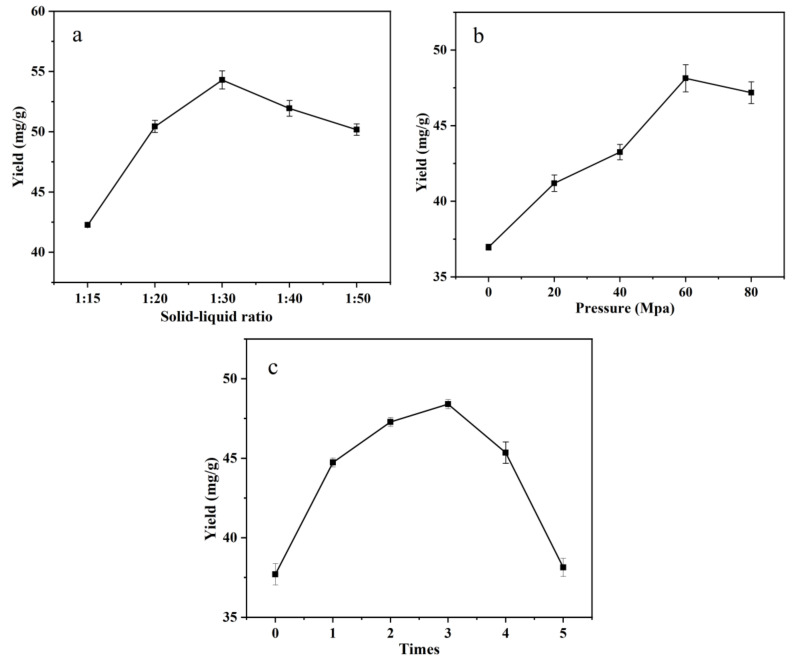
Single-factor experiments of variables’ mutual effect on IPS extraction. (**a**) Effect of solid-to-liquid ratio on IPS yield; (**b**) Effect of pressure on IPS yield; (**c**) Effect of times of extraction on IPS yield.

**Figure 3 foods-12-01067-f003:**
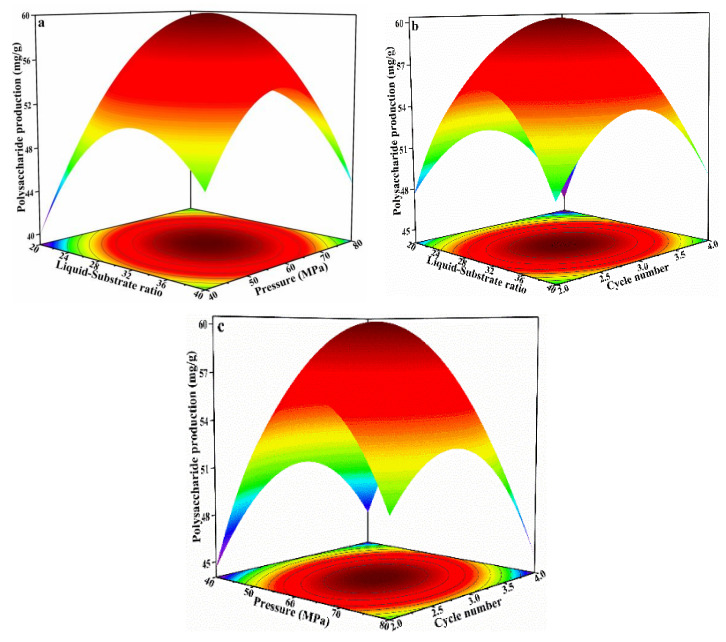
3D response profile of variables’ mutual effect on IPS extraction. (**a**) 3D response curve between liquid-substrate ratio and extraction pressure; (**b**) 3D response curve between liquid-substrate ratio and cycle number; (**c**) 3D response curve between pressure and cycle number.

**Figure 4 foods-12-01067-f004:**
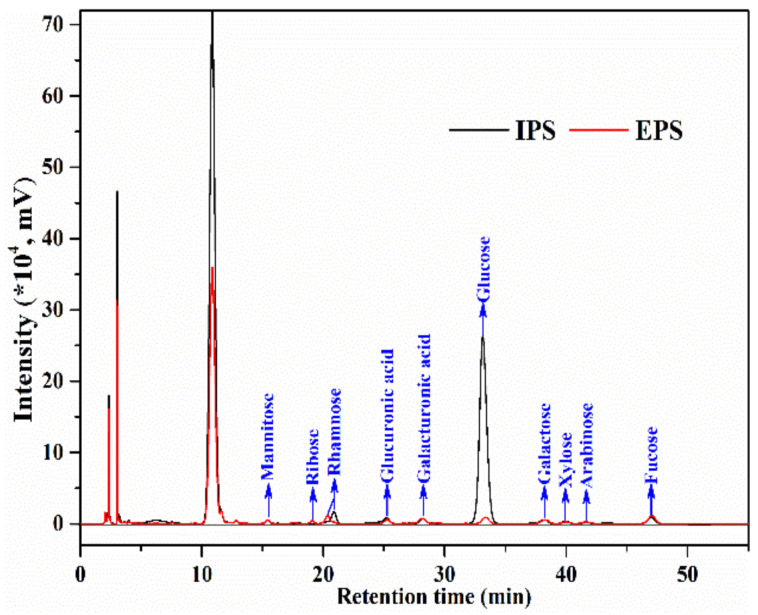
HPLC spectra of monosaccharide composition of IPS and EPS.

**Figure 5 foods-12-01067-f005:**
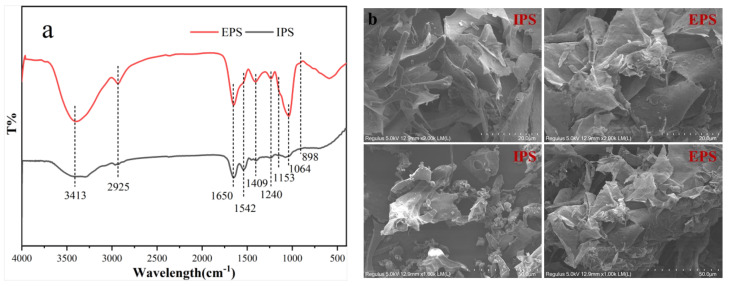
FTIR (**a**) and SEM (**b**) of EPS and IPS.

**Figure 6 foods-12-01067-f006:**
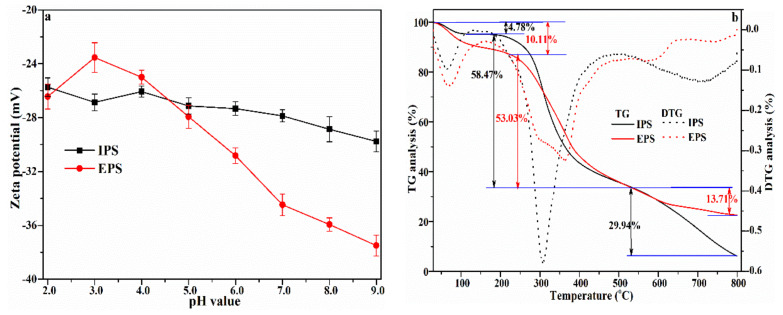
Zeta potential curves (**a**), TG/DTG curves of EPS and IPS (**b**).

**Figure 7 foods-12-01067-f007:**
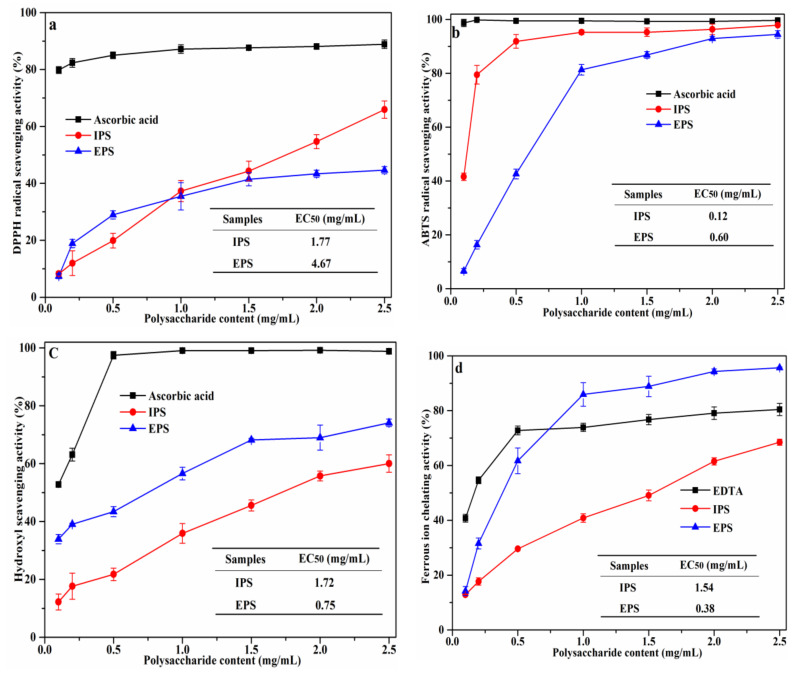
Antioxidant capacity of IPS and EPS: (**a**) DPPH radical; (**b**) ABTS radical; (**c**) HO· radical, (**d**) ferrous ion chelating.

**Table 1 foods-12-01067-t001:** Experimental design and results of RSM.

Number	Factors			Yield (mg/g)	
	A: Solid–Liquid Ratio	B: Pressure (Mpa)	C: Times	Observed	Predicted
1	1:20	40	3	39.87	39.88
2	1:30	60	3	60.33	60.12
3	1:30	60	3	60.33	60.12
4	1:30	40	4	46.60	46.23
5	1:30	60	3	60.40	60.12
6	1:40	60	4	48.77	48.77
7	1:20	80	3	46.58	46.14
8	1:20	60	4	44.63	45.06
9	1:40	80	3	44.47	44.45
10	1:30	80	2	49.24	49.68
11	1:40	40	3	46.35	46.79
12	1:30	60	3	60.04	60.12
13	1:40	60	2	49.59	49.16
14	1:30	80	4	45.07	45.07
15	1:30	40	2	44.60	44.59
16	1:20	60	2	47.65	47.64
17	1:30	60	3	59.54	60.12

**Table 2 foods-12-01067-t002:** ANOVA for RSM of IPS extraction.

Source	Sum of Square	df	Mean Square	F-Value	*p*-Value	Significant
Model	770.00	9	85.56	363.16	<0.0001	**
A	13.66	1	13.66	57.97	0.0001	**
B	7.70	1	7.70	32.67	0.0007	**
C	4.40	1	4.40	18.67	0.0035	**
AB	18.46	1	18.46	78.37	<0.0001	**
AC	1.21	1	1.21	5.51	0.0575	
BC	9.77	1	9.77	41.45	0.0004	**
A2	222.68	1	222.68	945.22	<0.0001	**
B2	306.89	1	306.89	1302.68	<0.0001	**
C2	113.59	1	113.59	482.15	<0.0001	**
Residual	1.65	7	0.2356			
Lack of Fit	1.15	3	0.3834	3.07	0.1532	
Pure Error	0.4990	4	0.1248			
R^2^	0.9979					
R^2^_adj_	0.9951					
Total variation	771.65	16				

Noting: ** *p* < 0.01.

**Table 3 foods-12-01067-t003:** Composition analysis of IPS and EPS.

Parameters	IPS	EPS	Parameters	IPS	EPS
Total sugar (%)	71.95	65.38	Glucuronic acid (%)	2.39	7.65
Protein (%)	3.71	4.41	Galacturonic acid (%)	1.88	11.13
Phenolic (%)	7.3	6.5	Glucose (%)	83.62	14.75
Ash (%)	5.7	7.8	Galactose (%)	1.77	10.78
Mannitol (%)	0.91	5.29	Xylose (%)	1.07	4.91
Ribose (%)	0.095	3.94	Arabinose (%)	0.60	5.96
Rhamnose (%)	4.42	15.61	Fucose (%)	3.25	19.99
Molecular weight (KDa)	272.85	185.13			

## Data Availability

Not applicable.

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
