# Peer review of "Preparation and Characterization of Intracellular and Exopolysaccharides during Cycle Cultivation of Spirulina platensis"

_foods, 2023, doi:10.3390/foods12051067_

Round 1

Reviewer 1 Report

The article is well structured but prior publications are some issues that need to be addressed. Please find my comments.

Comments :

Line 9: The the (remove one)

Line 71-75: Why did you reuse your medium for a second culture? And each time you want to reuse your medium?

There is something not clear in this part. The medium contains the sugars (polysaccharides) secreted by spirulina. And in your opinion, how will it become a medium containing sugars after five times of autoclave? Or it is reused without sterilization?

Line 84: for experimental design of RSM. How did you choose your variables and also the level of this variable?

Because before we do RSM, we should determine a significant single factor (preliminary study), and then we can choose the variable of the experiments.

Line 84: Extraction of IPS? IPS means intracellular polysaccharide, and in the protocol, you worked with the supernatant and precipitate with ethanol. Because the intracellular is something inside the cell, you should do lysis of the cell, and then you can have your IPS.

In this paragraph, I have the impression that you are talking about EPS, and the results confirm this you have the same results for IPS and EPS.

Line 72, 98: Please use italics for Latin names.

Line 99: Why did you filter your medium? I understand that you removed your cells by filtration because before the precipitation of EPS, you should remove your cells. And in this part, why did you not remove proteins?

Line 105 to 109: what is the interest of this part?

Line 145-146: For DPPH solution. You use pure ethanol for the precipitation of EPS and concentrated ethanol for antioxidant activity; my question is, when you mix your EPS with DPPH, your EPS is not precipitated. Why did you dissolve your DPPH in concentrated ethanol?

Because we use diluted ethanol for DPPH activity

Line 327: For the IPS, how did you know that the activity of the IPS and EPS is due to your polysaccharide and not expected for the phenolic compound?

Reviewer 2 Report

The manuscript is in general well structured and easy to read. However, some suggestion/remarks have been made about the interpretation of your results.

The suggestions/remarks have been added to the file in the attachment

The suggestion/remarks are added the attached file. 

Round 2

Reviewer 2 Report

Most of the comments/suggestions are adequately answered.